# Coronavirus Disease 2019 (COVID–19): A Short Review on Hematological Manifestations

**DOI:** 10.3390/pathogens9060493

**Published:** 2020-06-20

**Authors:** Artur Słomka, Mariusz Kowalewski, Ewa Żekanowska

**Affiliations:** 1Department of Pathophysiology, Nicolaus Copernicus University in Toruń, Ludwik Rydygier Collegium Medicum, 85–094 Bydgoszcz, Poland; zhemostazy@cm.umk.pl; 2Clinical Department of Cardiac Surgery, Central Clinical Hospital of the Ministry of Interior and Administration, Centre of Postgraduate Medical Education, 02–607 Warsaw, Poland; kowalewskimariusz@gazeta.pl; 3Department of Cardio–Thoracic Surgery, Heart and Vascular Centre, Maastricht University Medical Centre, 6229 Maastricht, The Netherlands; 4Thoracic Research Centre, Innovative Medical Forum, Nicolaus Copernicus University in Toruń, Ludwik Rydygier Collegium Medicum, 85–796 Bydgoszcz, Poland

**Keywords:** coronavirus disease 2019 COVID–19, severe acute respiratory syndrome coronavirus 2 SARS–CoV–2, hematology, hemostasis, cytokine storm

## Abstract

Infection with severe acute respiratory syndrome coronavirus 2 (SARS–CoV–2) is a rapidly spreading and devastating global pandemic. Many researchers are attempting to clarify the mechanisms of infection and to develop a drug or vaccine against the virus, but there are still no proven effective treatments. The present article reviews the common presenting hematological manifestations of coronavirus disease 2019 (COVID–19). Elucidating the changes in hematological parameters in SARS–CoV–2 infected patients could help to understand the pathophysiology of the disease and may provide early clues to diagnosis. Several studies have shown that hematological parameters are markers of disease severity and suggest that they mediate disease progression.

## 1. Introduction

### 1.1. Severe Acute Respiratory Syndrome Coronavirus 2 (SARS–CoV–2) at a Glance

The severe acute respiratory syndrome coronavirus 2 (SARS–CoV–2) contagion was first described in December 2019 when cases of pneumonia of an unidentified cause were reported in Wuhan, Hubei province, central China [1,2,3]. Occurring on 7 January 2020, Chinese scientists isolated a new type of coronavirus (CoV) [1] underlying this series of infections and, on 12 January 2020, reported its genetic sequence [1,3]. The source of the virus is currently understood to have been food products and wild animals that are easily available in the Huanan Seafood Wholesale Market, which was closed by the Chinese authorities on 1 January 2020 [1,2,3]. The infection has become a major global health concern and has attained pandemic status, having infected persons in almost every country worldwide [3,4].

To clarify the nomenclature of the newly identified coronavirus, the World Health Organization (WHO) proposed the designation “2019 novel coronavirus” (2019–nCoV) and the disease caused by it would be referred to as “coronavirus disease 2019” (COVID–19) [5]. The *Coronaviridae* Study Group (CSG) of the International Committee on Taxonomy of Viruses (ICTV) suggested that the virus be designated “SARS–CoV–2”, as it bears many similarities to severe acute respiratory syndrome (SARS)–causing coronaviruses (SARS–CoVs) [6]. Nonetheless, it has been suggested [5] that the name introduced by the CSG is inaccurate and may be misleading, as it suggests that the virus leads to the development of SARS, which led to the proposal of a new name: human coronavirus 2019 (HCoV–19) [5]. Here, we use the convention SARS–CoV–2 according to the proposal of the official association responsible for naming the *Coronaviridae* family [6].

SARS–CoV–2 is a member of the subgenus *Sarbecovirus* of the genus *Betacoronavirus* in the family *Coronaviridae* [7,8,9]. The family, with a large genome of 26–32 kilobases, belongs to a group of enveloped positive–sense single–stranded ribonucleic acid (RNA) viruses, referred to as (+)ssRNA viruses [10,11]. To date, several human CoVs have been identified, including OC43, 229E, NL63, HKU1, SARS–CoV and the Middle East respiratory syndrome coronavirus (MERS–CoV) [11,12,13]. The last two of the afore–mentioned viruses have been shown to have high infectivity and represented the etiological factors underlying the severe pneumonia epidemics in China [14] and Saudi Arabia [15]. Accumulating lines of evidence indicate that animals, particularly the bat species, may be pervasive reservoirs of coronaviruses [12,13].

Molecular and phylogenetic analyses of clinical samples from the respiratory tracts of infected patients have revealed that SARS–CoV–2 resembles two bat–borne SARS–like CoVs that had been previously isolated in China [7,8,9]. However, the assertion that animals are the exclusive source of infection is not entirely accurate, since the virus has spread rapidly to all continents globally [3] as a result of human-to-human transmission [16]. There is experimental evidence that human transmembrane angiotensin–converting enzyme 2 (ACE2) serves as a receptor for SARS–CoV–2 [17,18,19,20]. Briefly, the receptor–binding spike (S) glycoproteins, located on the surface of the virus, attach to the host cell through ACE2 and, consequently, the viral RNA may enter the host cell and replicate [18,19,20,21]. This process also is quite likely mediated by transmembrane serine protease 2 (TMPRSS2) [22]. The simplified structure of SARS–CoV–2 is shown in Figure 1.

Understanding the mechanism of interaction between virus and host cells is essential to the development of anti–SARS–CoV–2 drugs and vaccines [21].

### 1.2. Coronavirus Disease 2019 (COVID–19) at a Glance

Research concerning coronavirus disease 2019 (COVID–19) disease is in its infancy and the current global understanding of the pathophysiology, nosology, and symptomology of COVID–19 lacks both depth and universal agreement. Consequently, several diverse classifications have been proposed. Lai et al., [24] distinguished between three forms of COVID–19 manifestation, including (a) asymptomatic carrier; (b) acute respiratory disease (ARD) and (c) pneumonia of variable severity. Regarding the first group of patients, no clinical symptoms or radiographic changes are observed and most of these patients will not require hospitalization; however, they may still be a source of infection for other individuals [25,26,27]. A minor subgroup of hospitalized COVID–19 patients is diagnosed with ARD and experience typical respiratory symptoms without radiological signs of pneumonia [24]. The last minor subgroup of patients exhibits respiratory symptoms and radiological evidence of pneumonia, typically requiring hospitalization [24]. Yuki et al., [28] classified the disease manifestation into five separate categories, namely: (a) asymptomatic; (b) disease with mild symptoms (including fever, fatigue, myalgia, cough, sore throat, runny nose, sneezing, nausea, vomiting, abdominal pain and diarrhea); (c) moderate disease (pneumonia with frequent fever, cough without obvious hypoxemia and computed tomography (CT) of the chest showing lesions); (d) severe disease (pneumonia with hypoxemia and peripheral oxygen saturation, SpO_2_ < 92%); and (e) critical disease (associated with ARD, shock, encephalopathy, myocardial injury, heart failure, coagulation dysfunction and acute kidney injury). More recently, three main stages of the disease have been defined: stage I (mild symptoms observed) and stage II (pulmonary involvement detected) both lasting 5–7 days, with stage II being further divided into two substages, II A (no hypoxia) and II B (with hypoxia). The most severe stage, stage III (systemic inflammation) is attained by approximately 10–15% of patients [29].

Additionally, it has been suggested that patients having recovered from COVID–19 would have experienced biphasic disease: during the first phase, which lasts 7–10 days, worsening of clinical and radiological symptoms associated with intense virus replication would have been observed [30]. The second phase would have featured clinical and radiological improvement, accompanying reduction of viremia [30]. Furthermore, division of COVID–19 diagnoses into four categories, based primarily on the radiological results of the patients, has been proposed [31]. Despite these various classification criteria, most studies still only distinguish between mild and severe (intensive care requiring) disease manifestation, aiming to identify factors which may predict the severity of disease and be applicable to COVID–19 diagnosis and monitoring of treatment.

### 1.3. The Aim of the Present Review

The purpose of this short review is to familiarize readers with the primary hematological manifestations of severe acute respiratory syndrome coronavirus 2 (SARS–CoV–2) infection. Although diagnosis of coronavirus disease 2019 (COVID–19) is challenging in the early stages due to non–obvious manifestations, hematological signs and symptoms provide clues to aid diagnosis. We separately address the effects on white blood cells (WBCs), red blood cells (RBCs), platelets (PLTs) and coagulation.

## 2. Hematologic Symptoms of Severe Acute Respiratory Syndrome Coronavirus 2 (SARS–CoV–2) Infection

### 2.1. White Blood Cells (WBCs)

Laboratory–confirmed severe acute respiratory syndrome coronavirus 2 (SARS–CoV–2) infection is associated with alterations in the white blood cell (WBC) count. One in four COVID–19 positive patients experience some form of leukopenia (WBC < 4 × 10⁹ cells/L), with the majority (63.0%) exhibiting lymphocytopenia (lymphocyte count < 1 × 10⁹ cells/L) [32]. One study reported 45.0% of patients to have WBC counts within the normal range (4–10 × 10^9^ cells/L), while 30.0% had an elevated WBC count (> 10 × 10⁹ cells/L) [32]. The extent of deviation from normal WBC counts appeared to correlate with disease severity, as patients with severe disease were found to have two–folds higher WBC counts than those with non–severe disease [32]. Furthermore, blood from severely ill patients featured more neutrophils and fewer lymphocytes than blood from patients with non–severe disease [32]. Another study, published in JAMA: The Journal of the American Medical Association, found that, in patients hospitalized with coronavirus disease 2019 (COVID–19), those with severe symptoms had an elevated WBC and neutrophil count, as well as a lower lymphocyte count (median of 0.8 × 10⁹ cells/L, interquartile range 0.5–0.9 × 10⁹ cells/L) than patients with non–severe disease manifestation (median of 0.9 × 10⁹/L, interquartile range 0.6–1.2 × 10⁹ cells/L) [33]. Patients who did not survive the disease had exhibited, during hospitalization, more advanced lymphocytopenia than patients who recovered [33]. The importance of a low lymphocyte count as a hematological symptom of COVID–19 infection has become evident from several studies, including that of Lui et al., [34] who showed that the majority of patients (72.3%) had a lymphocyte count below 1 × 10⁹ cells/L of blood. A higher proportion of patients with severe symptoms (61.1%) being diagnosed with leukopenia, as opposed to the general 33.7% of patients across all disease severity levels who are diagnosed with leukopenia [35]. Furthermore, the severely ill subgroup also featured lymphocytopenia in 96.1% of patients, which confirmed previous observations [35].

The data described thus far predominantly represented patients from Wuhan, China. To contrast, some studies involving younger patients from outside of Wuhan reported no marked change in WBC counts, a lower severity of symptoms and a full recovery, particularly in adults with no co–morbidities [36]. Another study conducted outside of Wuhan reported normal WBC counts in 68.0% of patients and lymphocytopenia diagnoses in 42.0% of patients, most of whom exhibited only mild to moderate clinical symptoms [37]. Therefore, it would appear the age of the patient and the stage of the disease may be key factors which can determine the presence of lymphocytopenia and its progression in patients positive for COVID–19. Added to lymphocytopenia, a reduced number of eosinophils, or eosinopenia, has been reported in more than half (52.9–78.8%) of patients who tested positive for COVID–19 [38,39]. Table 1 shows a comparison of the number of WBCs, lymphocytes, neutrophils, monocytes, and eosinophils between patients with non–severe and severe COVID–19.

Moreover, clinically important observations regarding hematological changes during COVID–19 disease progression are provided by the microscopic analysis of the peripheral blood smears. Untreated COVID–19 patients exhibited accelerated and disordered granulopoiesis [40]. Concerning blood smears, very rare anomalies such as leukoerythroblastic reaction [41] and blue–green leukocyte inclusions also have been observed [42]. Singh et al., [43] also suggested that the appearance of activated monocytes in a blood smear may indicate an improvement in the patient’s clinical condition.

The possible role of lymphocyte subpopulations in the host response to SARS–CoV–2 infection has been the subject of several pilot studies, several of which reported below–normal numbers of CD3^+^, CD4^+^ and CD8^+^ T cells detected upon admission, as well as one week after admission, in patients with severe disease manifestation [44,45,46]. The expression of interferon gamma (IFN–γ) by CD4^+^T cells, which may be crucial in antiviral responses, was shown to be lower in seriously ill patients [44]. It has been further suggested that the neutrophil–to–CD8^+^ T cell ratio (N8R) bears potential as a diagnostic parameter for severe disease manifestation, with an area under the curve (AUC) of 0.94 [45]. Recent publications [47,48,49,50] have suggested a high prognostic value in the number of circulating neutrophils and lymphocytes, expressed as the neutrophil to lymphocyte ratio (NLR), for the prediction of disease severity, with a high NLR being associated with the severe manifestation of COVID–19 disease. Another indicator that can be employed to evaluate the severity of disease is the lymphocyte to C-reactive protein ratio (LCR), which was markedly reduced in cases of severe COVID–19 disease [50]. Although these data are fragmented and remain to be further investigated, the attenuation of the adaptive immune system may be a feature and one of the laboratory–detectable parameters of the disease.

To summarize, from the analysis of the available literature, it is evident that the majority of patients (especially those with severe disease, residing in intensive care units) are likely to develop lymphocytopenia. This laboratory symptom applies primarily to adult patients rather than children [51] and can predict disease severity [52,53]. It should be noted that the reduced number of lymphocytes is also a characteristic of diseases caused by other coronaviruses primarily infecting the human respiratory tract, including severe acute respiratory syndrome (SARS) coronavirus [54] and Middle East respiratory syndrome (MERS) coronavirus [55]. The mechanism of this phenomenon is not fully understood, although some authors hypothesize that it is associated with the intensification of the inflammatory process (cytokine storm syndrome) and/or direct infection of lymphocytes and destruction of lymphoid organs [52,56]. These hypothetical considerations are now supported by the observations that the virus can infect T cells through receptor–dependent, S protein–mediated membrane fusion [57], which results in depletion of the cytotoxic capacity of lymphocytes [58].

Additionally, initial treatments administered to patients with moderate to severe respiratory symptoms, especially glucocorticosteroids, may themselves lead to lymphocytopenia [59]. It also should be noted that there is currently insufficient clinical evidence to indicate the use of this group of drugs for the treatment of COVID–19 patients [60]. Considering the high risk of side effects of glucocorticosteroids use, some authors have recommended not using them in hospital practice [61,62], unless a patient’s clinical condition necessitates it. The mechanisms leading to a decrease in the lymphocyte count during COVID–19, as well as the clinical significance of this process, warrant further intensive research.

### 2.2. Red Blood Cells (RBCs)

Most publications to date show that the red blood cell (RBC) system in patients with coronavirus disease 2019 (COVID–19) is not affected [32,35,36,37,39,44,45,46,47]. The most frequent laboratory parameter reported by authors is hemoglobin, while hematocrit [36] and the number of RBCs were less frequently reported [39]. Analyses performed by several authors showed no differences between the hemoglobin levels of patients with severe COVID–19 symptoms and those with a mild/moderate disease [32,39,44,45,46,47]. However, Lippi and Mattiuzzi [64] conducted a meta–analysis of four studies and showed that patients with severe symptoms exhibited lower hemoglobin levels compared to those with mild symptoms. These findings should be interpreted conservatively, as these studies included a small number of patients (*n =* 1210) and exhibited much heterogeneity between the studies (81.0%).

The RBC–related laboratory parameter ferritin is routinely used to assess iron metabolism. It has been noted that patients with severe COVID–19 also appear to have increased ferritin levels [44,45]. This may be as a result of inflammation, as ferritin is a positive acute phase protein [65]. COVID–19 has many features in common with classic hyperferritinemic syndromes and, for this reason, also could be classified along with this group of diseases which also feature severe inflammation, lymphocytopenia, abnormal liver function tests and coagulopathy [66].

Several studies [63,67,68] reported the erythrocyte sedimentation rate (ESR) is significantly higher in patients with severe disease, which has been confirmed by a recent meta–analysis of three studies, including a total of 819 patients [69]. Surprisingly, Zhang et al., [67] demonstrated that ESR has proven to be a powerful parameter for predicting the severity of COVID–19 (AUC = 0.951).

Last, patients with blood group A appear to be more prone to infection, however, the mechanisms responsible for this effect are unknown [70,71]. However, both papers which make this correlation are as yet published only as preprints, hence the authors’ interpretation of the results should be considered tentative until completion of the peer review process. Table 2 shows a comparison of the RBC–related laboratory parameters between patients with non–severe and severe COVID–19.

### 2.3. Platelets (PLTs)

Platelets are an important element of hemostasis and are involved in many physiological and pathological processes, including an important modulation of inflammatory responses [72]. Most of the papers reviewed here did not indicate the presence of thrombocytopenia or platelet differences between patients with severe disease and those exhibiting mild disease [32,33,44,45,46,47]. One paper involving the largest study group reviewed (*n* = 1099) showed a reduced platelet count in more than half of the patients (57.5%) in the intensive care unit, with a median count of 137,500 platelets/µL (interquartile range 99,000–179,500 platelets/µL) [35]. Despite the differences in individual observations made by the authors, it may be advisable that the platelet count, and the dynamics of changes in these counts, be monitored in each patient. This aligns with the meta–analytical data of Lippi et al., [73] which suggested that a low platelet count is associated with a severe form of the disease. Nonetheless, as in the previous meta–analysis regarding hemoglobin levels [64], these results also must be interpreted conservatively, as the heterogeneity among the nine included studies was 92.0% [73]. Although patients with a severe disease may be at a risk of developing thrombocytopenia [74], the mechanism of this condition remains obscure. It has been postulated that thrombocytopenia may result from three predominant mechanisms, including decreased platelet production, as well as increased destruction and increased consumption of platelets. However, these are theoretical postulations based on experimental and clinical observations of the course of other viral infections [75,76] and further research is recommended regarding the potential mechanism of thrombocytopenia and the role of platelets in COVID–19.

### 2.4. Plasma Hemostatic Parameters

Alterations in D–dimer levels are the most commonly observed anomaly of the hemostasis system in patients with coronavirus disease 2019 (COVID–19) [77]. Most researchers have reported a marked increase in the levels of D–dimer in patients with severe disease [32,34,35,38,44,45,67], which also was reported in a pooled analysis of four studies [78]. Elevated D–dimer levels are associated with high levels of fibrin degradation products (FDPs) and low antithrombin (AT) activity [79,80]. Levels of other laboratory parameters evaluated for the routine assessment of blood coagulation appear to be normal, regardless of the severity of infection, except for reduced activated partial thromboplastin time (aPTT) [44] in severe SARS–CoV–2–infected cases and a single case of prolonged prothrombin time (PT) [32]. Table 3 shows a comparison of the number of platelets and plasma hemostatic parameters between patients with non–severe and severe COVID–19.

D–dimers are a sensitive and specific marker of coagulation and fibrinolysis activation, as well as being crucial to the diagnosis of disseminated intravascular coagulation (DIC) [81,82]. Patients with viral infection, as well as COVID–19 disease, are at high risk of developing this complication [80,83] and should be carefully monitored for possible development of thrombotic and hemorrhagic complications. During the pathophysiology of thrombotic complications associated with SARS–CoV–2 infection, a crucial role is played by endothelial dysfunctions [84,85,86,87,88]. The levels and activity of the von Willebrand factor (vWF) and factor VIII were elevated in severe COVID–19 cases [84,85,88]. Additionally, other markers of endothelial dysfunctions, including increased soluble fms–like tyrosine kinase 1/placental growth factor (sFLT1/PlGF) ratios, and the presence of viral elements within endothelial cells have been reported [86,87]. Blood obtained from patients with COVID–19 has been found to be characterized by hyperviscosity, which further increases the risk of endothelium damage and thrombosis [89].

Studies implementing viscoelastic hemostasis assays (thromboelastometry and thromboelastography) have provided valuable insights into blood coagulation imbalances observed in patients with COVID–19. Results have consistently shown that severe and critically ill COVID–19 patients frequently develop a hypercoagulability, which may persist over time [88,90,91,92]. Simultaneously, signs of systemic fibrinolysis were not found in laboratory analyses [88,90,91,92]. Clinical observations [93,94,95,96,97,98], though conducted on relatively small groups of patients, revealed a high rate of venous and arterial thrombotic events, strongly emphasizing the urgent need to include a thromboprophylaxis and anticoagulant regimen in the treatment of COVID–19 [99,100].

International and national medical societies, including the International Society on Thrombosis and Haemostasis (ISTH) [101], the International Federation of Clinical Chemistry and Laboratory Medicine (IFCC) [102], the European Society of Cardiology (ESC) [103], the Italian Society on Thrombosis and Haemostasis (SISET) [104] and the Polish Association of Epidemiologists and Infectiologists (PTEiLChZ) [105] recommend that COVID–19 patients should be monitored primarily for D–dimer levels, as well as PT and PLT counts. Laboratory medicine and laboratory hematology currently, and will continue to, play a role in the diagnosis and monitoring of the progression and treatment of this novel viral disease [106,107,108]. The literature also indicates a possible role for drugs commonly used in hematological diseases, repurposed for COVID–19 therapy [109].

## 3. Cytokine Storm—A Link between Inflammation and Thrombosis in Coronavirus Disease 2019 (COVID–19)

One of the most potent factors in the pathophysiology of coronavirus disease 2019 (COVID–19) is the cytokine storm [110], defined as a clinical state of hyperinflammation due to over–biosynthesis and the release of proinflammatory cytokines and chemokines [111,112]. Cytokine storm syndrome may result in multiorgan failure, primarily due to extensive local edema, which can be fatal [113,114]. The cytokine/chemokine release underlying a cytokine storm has powerful proinflammatory properties. High levels of these molecules, including interleukin (IL) 6, IL–2, IL–2 receptor (IL–2R), IL–10, tumor necrosis factor alpha (TNF–α) and interferon gamma (IFN–γ), have been described in patients with severe and refractory COVID–19 [44,45,63,115]. Similarly, patients who died of the disease also exhibited elevated synthesis of IL–6, IL–2R, IL–8, IL–10 and TNF–α in comparison to recovered patients [116]. Moreover, two recent meta–analyses substantiated the evidence of high IL–6 levels in severe disease cases [117] and demonstrated that high IL–6 levels at the time of hospital admission may be associated with high COVID–19 mortality [118]. Based on the frequency at which patients with severe COVID–19 suffer a cytokine storm, the use of cytokine–blocking drugs has been proposed in COVID–19 disease therapy, including anti–TNF antibodies [119], the IL–6 receptor (IL–6R) antagonist tocilizumab [120,121] and the IL–1R antagonist anakinra [122,123]. The pathophysiology of COVID–19 combines an intense inflammatory process with a disorder of hemostasis in the form of hypercoagulable states, involving both microangiopathy and systemic coagulation defects [110,124]. Thromboinflammation thus represents a compound factor in the development and progression of COVID–19 disease, as several components of both processes, inflammation and thrombosis, are common and overlapping [110,125,126,127].

## 4. Coronavirus Disease 2019 (COVID–19) in Relation to Sex—A Lack of Data on the Relationship between Hematological Changes

Preliminary studies have suggested that the sex of an individual is a factor associated with the risk of severe acute respiratory syndrome coronavirus 2 (SARS–CoV–2) infection and the severity of the disease [128,129,130]. De Lusignan et al., [128] found that the male sex is associated with an increased risk of SARS–CoV–2 infection. Additionally, males often develop a more severe form of coronavirus disease 2019 (COVID–19) and are more likely to die from the disease compared with females [129,130]. Dangis et al., [131] drew a computed tomography (CT)–based association between the male sex and extensive lung disease, independently of age and time of symptoms onset. There are several potential explanations for these observations, primarily related to the higher incidence of comorbidities and a higher likelihood of risk behaviors, as well as elevated soluble angiotensin–converting enzyme 2 (sACE2) levels in males [132,133]. Presently, there are no analyses that compare laboratory hematological parameters according to the sex of patients with COVID–19. Such observations may provide valuable data regarding the possible etiology of severe disease in males, therefore providing further clues to improve treatment strategies for COVID–19. Extremely valuable in practical terms would be the observation whether lymphocytopenia or high D–dimer levels differentially occur in males versus females, alluding to their significance in predicting the severity of the disease and the effectiveness of treatment.

## 5. Prediction of Severity and Mortality of Coronavirus Disease 2019 (COVID–19)—A Summary on Hematological Changes

Described in previous chapters, parameters exposing changes in the hematological system appear to be highly valuable in predicting coronavirus disease 2019 (COVID–19) severity and the risk of patient mortality. Furthermore, their use in everyday clinical practice is comparatively feasible. Somewhat outdated parameters such as erythrocyte sedimentation rate (ESR) may be undergoing a revival as detection methods for excessive inflammation [134]. Neutrophil to lymphocyte ratio (NLR) and plasma D–dimer levels are also relatively easy to determine and of great prognostic value. Based on their feasibility and value, the inclusion of these tests in clinical practice is likely to accelerate. To consolidate the interpretation of the parameter values described in the subsections of this review, these data are summarized in Table 4 below.

## 6. Conclusions and Further Directions

Coronavirus disease 2019 (COVID–19) is a rapidly evolving viral disorder and early stage diagnosis poses a great challenge to physicians. Insights drawn from studies to date provide compounding evidence regarding the significance of leukocytosis, lymphocytopenia and increased D–dimer levels in the prognosis facing severe acute respiratory syndrome coronavirus 2 (SARS–CoV–2) infected patients (Figure 2).

Despite progress in the elucidation of COVID–19 disease mechanisms, many aspects remain unclear and continue to present an imposing challenge to the scientific community [135]. Taking a pathophysiological point of view, the mechanisms responsible for the infection and deterioration of the patient’s clinical condition are still poorly understood. This particularly concerns the immune response [136], including the role of cells such as monocytes/macrophages [137] and other cytokines such as interleukin 17A (IL–17A) [138]. The prevalence of thrombocytopenia [139] and the absence of laboratory signs of fibrinolysis in viscoelastic methods emphasize the importance of blood coagulation in the course of acute COVID–19. Furthermore, the long–term effects of infection on the hematological system are not evident yet. This will facilitate a deeper understanding of the role of the hematological system in the disease pathophysiology and progression. Taking a clinical point of view, development of effective treatment and vaccine regimens is highly dependent on a greater understanding of the pathomechanisms of COVID–19 and its associated complications. Due to the novelty of COVID–19 as a pandemic–level disease, many studies offer preliminary findings (often in pre–print form) and presently lack longitudinal and cross–sectional data, justifying the tentative interpretation of their conclusions [140].

## Figures and Tables

**Figure 1 pathogens-09-00493-f001:**
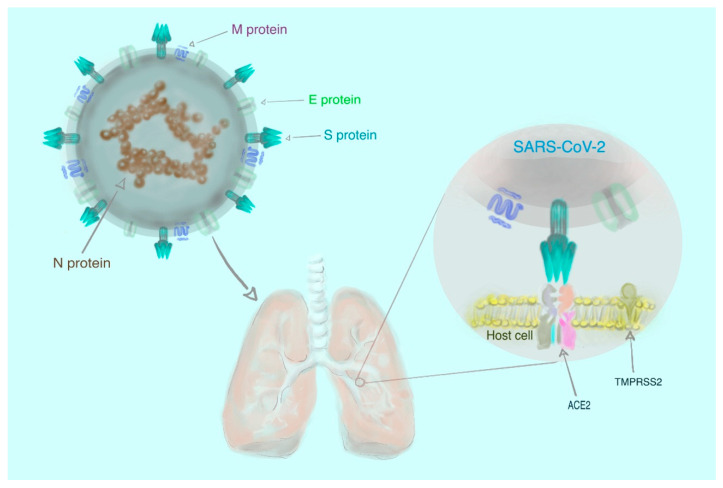
The structure of severe acute respiratory syndrome coronavirus 2 (SARS–CoV–2) comprising four different proteins: spike (S), membrane (M), envelope (E) and nucleocapsid (N). The first three proteins (S, M, and E) are the components of the viral envelope, while the N protein (enclosing RNA) forms a part of the genome. To enter the cell, the virus requires the interaction of the S protein with human transmembrane angiotensin–converting enzyme 2 (ACE2) and transmembrane serine protease 2 (TMPRSS2) [22,23].

**Figure 2 pathogens-09-00493-f002:**
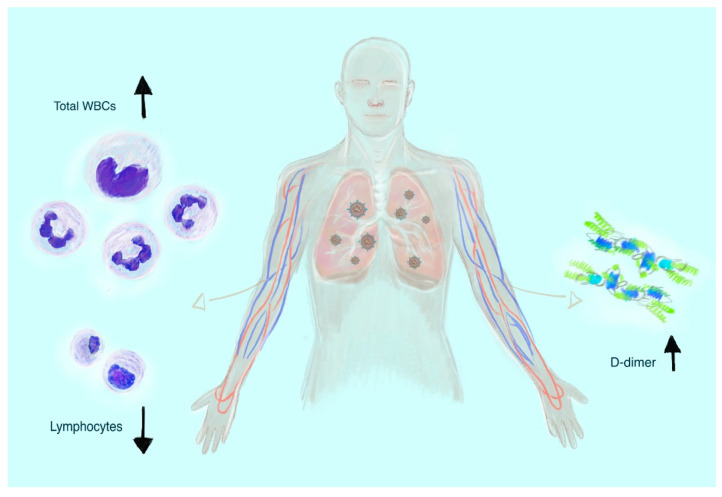
The most common hematological laboratory symptoms of severe coronavirus disease 2019 (COVID–19) manifestation: leukocytosis, lymphocytopenia, and increased D–dimer levels.

**Table 1 pathogens-09-00493-t001:** Comparison of the number of white blood cells (WBCs), lymphocytes, neutrophils, monocytes, and eosinophils between patients with non–severe and severe coronavirus disease 2019 (COVID–19).

Parameter [Unit]		Non–Severe COVID–19	Severe COVID–19	Probability Value (Non–Severe versus Severe COVID–19)	Reference
White blood cells (WBCs)	WBC count [×10^9^/L]	5.7 (3.1–7.6) *n* = 28	11.3 (5.8–12.1) *n* = 13	0.001	[32]
		4.3 (3.3–5.4) *n =* 102	6.6 (3.6–9.8) *n =* 36	0.003	[33]
		4.9 (3.8–6.0) *n =* 926	3.7 (3.0–6.2) *n =* 173	not determined	[35]
		4.5 (3.5–5.9) *n =* 82	5.3 (4.0–9.0) *n =* 56	0.014	[38]
		4.5 (3.9–5.5) *n =* 10	8.3 (6.2–10.4) *n =* 11	0.003	[44]
		3.9 ± 1.5 *n =* 27	6.6 ± 3.4 *n =* 13	not determined	[45]
		4.7 (4.0–5.8) *n =* 58	5.1 (3.5–8.2) *n =* 9	0.87	[46]
		6.4 ± 2.4 *n =* 69	9.1 ± 5.6 *n =* 24	0.006	[49]
		4.5 (3.5–5.5) *n =* 240	4.5 (3.7–6.2) *n=* 58	0.442	[63]
	Lymphocyte count [×10^9^/L]	1.0 (0.7–1.1) *n =* 28	0.4 (0.2–0.8) *n =* 13	0.0041	[32]
		0.9 (0.6–1.2) *n =* 102	0.8 (0.5–0.9) *n =* 36	0.03	[33]
		1.0 (0.8–1.4) *n =* 926	0.8 (0.6–1.0) *n =* 173	not determined	[35]
		0.8 (0.6–1.2) *n =* 82	0.7 (0.5–1.0) *n =* 56	0.048	[38]
		1.1 (1.0–1.2) *n =* 10	0.7 (0.5–0.9) *n =* 11	0.049	[44]
		1.1 (0.8–1.4) *n =* 27	0.6 (0.6–0.8) *n =* 13	not determined	[45]
		1.3 (0.9–1.7) *n =* 58	0.5 (0.48–0.8) *n =* 9	0.0002	[46]
		1.17 ± 0.63 *n =* 69	0.65 ± 0.54 *n =* 24	< 0.001	[49]
		1.3 (1.0–1.8) *n =* 240	0.9 (0.7–1.2) *n =* 58	< 0.001	[63]
	Neutrophil count [×10^9^/L]	4.4 (2.0–6.1) *n =* 28	10.6 (5.0–11.8) *n =* 13	0.00069	[32]
		2.7 (1.9–3.9) *n =* 102	4.6 (2.6–7.9) *n =* 36	< 0.001	[33]
		2.7 (2.1–3.7) *n =* 10	6.9 (4.9–9.1) *n =* 11	0.002	[44]
		2.0 (1.5–2.9) *n =* 27	4.7 (3.6–5.8) *n =* 13	not determined	[45]
		2.6 (2.1–3.8) *n =* 58	4.2 (2.1–6.9) *n =* 9	0.17	[46]
		4.55 ± 0.21 *n =* 69	7.73 ± 5.4 *n =* 24	< 0.001	[49]
		6.6 (5.3–8.7) *n =* 240	7.3 (5.4–9.6) *n =* 58	0.158	[63]
	Monocyte count [×10^9^/L]	0.4 (0.3–0.5) *n =* 102	0.4 (0.3–0.5) *n =* 36	0.96	[33]
		0.3 (0.2–0.5) *n =* 27	0.2 (0.2–0.5) *n =* 13	not determined	[45]
		0.5 (0.4–0.6) *n =* 58	0.3 (0.2–0.5) *n =* 9	0.12	[46]
		0.41 ± 0.2 *n =* 69	0.5 ± 0.84 *n =* 24	0.045	[49]
	Eosinophil count [×10^9^/L]	0.02 (0.008–0.05) *n =* 82	0.01 (0.0–0.06) *n =* 56	0.451	[38]
		0.02 (0–0.05) *n =* 240	0.01 (0–0.03) *n =* 58	< 0.001	[63]

**Table 2 pathogens-09-00493-t002:** Comparison of the red blood cell (RBC)–related laboratory parameters between patients with non–severe and severe coronavirus disease 2019 (COVID–19).

Parameter [Unit]		Non–Severe COVID–19	Severe COVID–19	Probability Value (Non–Severe versus Severe COVID–19)	Reference
Red blood cell (RBC)–related parameters	Hemoglobin levels [g/L]	130.5 (120.0–140.0) *n =* 28	122.0 (111.0–128.0) *n* = 13	0.20	[32]
		135 (120.0–148.0) *n =* 926	128.0 (112.0–141.0) *n =* 173	not determined	[35]
		139.5 (132.8–146.0) *n =* 10	136.0 (125.5–144.5) *n =* 11	0.78	[44]
		127.8 ± 13.1 *n =* 27	123.4 ± 14.0 *n =* 13	not determined	[45]
		142.0 (129.0–152.0) *n =* 58	132.0 (125.0–140.0) *n =* 9	0.07	[46]
	Ferritin levels [µg/L]	337.4 (286.2–1275.4) *n =* 10	1598.2 (1424.6–2036.0) *n =* 11	0.049	[44]
		367.8 (174.7–522.0) *n =* 27	835.5 (635.4–1538.8) *n =* 13	not determined	[45]
	Erythrocyte sedimentation rate; ESR [mm/h]	24.0 (13.5–42.5) *n =* 240	45.0 (28.0–61.0) *n =* 58	< 0.001	[63]

**Table 3 pathogens-09-00493-t003:** Comparison of the number of platelets (PLTs) and plasma hemostatic parameters between patients with non–severe and severe coronavirus disease 2019 (COVID–19).

Parameter [Unit]	Non–Severe COVID–19	Severe COVID–19	Probability Value (Non–Severe versus Severe COVID–19)	Reference
Platelet (PLT) count [× 10^9^/L]	149.0 (131.0–263.0) *n =* 28	196.0 (165.0–263.0) *n =* 13	0.45	[32]
	165.0 (125.0–188.0) *n =* 102	142.0 (119.0–202.0) *n =* 36	0.78	[33]
	172.0 (139.0–212.0) *n =* 926	137.5 (99.0–179.5) *n =* 173	not determined	[35]
	175.6 (148.3–194.0) *n =* 10	157.0 (134.0–184.5) *n =* 11	0.88	[44]
	181.4 ± 70.7 *n =* 27	186.6 ± 68.1 *n =* 13	not determined	[45]
	201.0 (157.0–263.0) *n =* 58	217.0 (154.0–301.0) *n =* 9	0.81	[46]
Prothrombin time; PT [s]	10.7 (9.8–12.1) *n =* 28	12.2 (11.2–13.4) *n =* 13	0.012	[32]
	12.9 (12.3–13.4) *n =* 102	13.2 (12.3–14.5) *n =* 36	0.37	[33]
	13.4 (12.8–13.7) *n =* 10	14.3 (13.6–14.6) *n =* 11	0.15	[44]
	13.1 ± 0.6 *n =* 27	13.4 ± 0.6 *n =* 13	not determined	[45]
International normalized ratio (INR)	1.0 ± 0.1 *n =* 27	1.0 ± 0.1 *n =* 13	not determined	[45]
Activated partial thromboplastin time; aPTT [s]	27.7 (24.8–34.1) *n =* 28	26.2 (22.5–33.9) *n =* 13	0.57	[32]
	31.7 (29.6–33.5) *n =* 102	30.4 (28.0–33.5) *n =* 36	0.09	[33]
	44.0 (42.6–47.6) *n =* 10	33.7 (32.1–38.4) *n =* 11	0.002	[44]
	39.5 ± 4.6 *n =* 27	39.5 ± 4.2 *n =* 13	not determined	[45]
Fibrinogen levels [g/L]	4.5 ± 1.4 *n =* 27	6.3 ± 1.3 *n =* 13	not determined	[45]
D–dimer levels [mg/L]	0.5 (0.3–0.8) *n =* 28	2.4 (0.6–14.4) *n =* 13	0.0042	[32]
	166 (101–285) *n =* 102	414 (191–1324) *n =* 36	< 0.001	[33]
	0.2 (0.1–0.3) *n =* 82	0.4 (0.2–2.4) *n =* 56	< 0.001	[38]
	0.3 (0.3–0.4) *n =* 10	2.6 (0.6–18.7) *n =* 11	0.029	[44]
	0.4 (0.2–0.8) *n =* 27	0.9 (0.7–1.5) *n =* 13	not determined	[45]
	0.3 (0.2–0.5) *n =* 240	0.5 (0.3–0.9) *n =* 58	< 0.001	[63]

**Table 4 pathogens-09-00493-t004:** The clinical value of hematological parameters for prognosis of coronavirus disease 2019 (COVID–19) patients.

Parameter		Clinical Value	Reference
White blood cell (WBC) –related parameters	WBC count	↑ in severe cases	[32,33,38,44,45,49]
	Lymphocyte count	↓ in severe cases Early prognosis of severity	[32,33,35,38,44,45,46,49,52,53,63]
	CD3^+^, CD4^+^, CD8^+^ T cell count	↓ in severe patients	[44,45,46,67]
	Neutrophil to CD8^+^ T cell ratio (N8R)	↑ in severe cases Early prognosis of severity	[45,67]
	Neutrophil to lymphocyte ratio (NLR)	↑ in severe cases Early prognosis of severity	[45,47,48,49,50,67]
	Lymphocyte to C–reactive protein ratio (LCR)	↓ in severe cases	[50]
Red blood cell (RBC) –related parameters	Ferritin levels	↑ in severe cases	[44,45]
	Erythrocyte sedimentation rate (ESR)	↑ in severe cases Early prognosis of severity	[63,67,68,69]
Platelet (PLT) count		↓ in severe patients Early prognosis of severity and mortality	[35,73]
D–dimer levels		↑ in severe cases	[32,33,38,44,45,63]

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
