# Peer review of "Coronavirus Disease 2019 (COVID–19): A Short Review on Hematological Manifestations"

_pathogens, 2020, doi:10.3390/pathogens9060493_

Round 1

Reviewer 1 Report

Manuscript is not carefully organized and presented. I suggest authors to include a section called "introduction" which the aim of the proposed review is clearly reported with a comprehensive background properly referenced. For example introduction divided into sub-sections. Moreover, sometimes there there are different typos (e.g. L167-168, 195-199). English need to be checked.

In different cases, authors referenced sentences by multiples sources reporting data just from a single reference (e.g. L102-105 in which authors use data from [32-35] reporting only data retrieved from [32]. It would be useful for the reader, especially for this particular topic to have data clearly explained. The use of Tables is encouraged of all analytes considered.  

in L105-106 authors refer to a study but they do not provide reference.

L113 authors report the median of lymphocyte count. It should be important to add the IQR for all data, where provided.

Please check the percentage reported in L 126.

I suggest to use use uniform nomeclature for the virus (Sars-CoV-2).

Even though most of the data are available from few days. I encourage the use of peer-reviewed papers instead of manuscript uploaded on preprint server.

On L173, authors should refer to a ROC curve for the reported AUC.

Please check if all the acronyms are defined the first time they appear in the manuscript. Moreover, i would use the extended names when reported as titles of sub-headings.

One of the weakness of the manuscript is that data are reported carefully, but a reader would like to have a critical and in depth discussion of the data reported in literature.

I would suggest to add data about Sars-CoV-2 vs. the gender.

Please check the reference used in L181-183. It seems not properly referenced. If yes, i suggest authors to go through the references in order to reduce inconsistencies.

No data about cytokines is reported. I suggest to add a part related to at least IL-6.

Conclusions are not properly organized and is not of worth to place a Table and a figure here.  Table 1 can have interest if properly placed and expanded.

Table caption and Conclusions are mixed together (L216-227).

Author Response

Bydgoszcz, June, 12 2020

Dr. Artur SĹ‚omka

Department of Pathophysiology

Ludwik Rydygier Collegium Medicum in Bydgoszcz

Nicolaus Copernicus University in Toruń

Prof. Dr. Lawrence S. Young

Editor-in-Chief

Pathogens

Dear Prof. Young,

Thank you very much for your correspondence dated on 2020-05-23 regarding our manuscript authored by Artur SĹ‚omka et al., entitled “Coronavirus disease 2019 (COVID-19): a short review on hematological  manifestations” (pathogens-818772), submitted for publication in Pathogens. We are grateful that you consider our manuscript as likely suitable for publication. We have carefully rewritten the manuscript taking advantage of the Reviewer comments.

Enclosed please find our detailed responses to specific remarks of the Referees.

Yours sincerely,

Dr. Artur SĹ‚omka

REPLY TO REVIEWER #1 COMMENTS

We appreciate your interest in our manuscript and the most valuable suggestions to improve its quality. All the changes are highlighted in yellow in the revised text.

Below, we enclose a point-by-point response to your comments.

  1. Manuscript is not carefully organized and presented. I suggest authors to include a section called “introduction” which the aim of the proposed review is clearly reported with a comprehensive background properly referenced. For example introduction divided into sub-sections. Moreover, sometimes there are different typos (e.g. L167-168, 195-199). English need to be checked.

Reply: Thank you very much for your comments. As per the Reviewer’s suggestion, the introduction has been divided into subsections. All typos have been corrected, thank you. English was checked by a native English speaker and effort was made to propose a more structured version of our paper.

  1. In different cases, authors referenced sentences by multiples sources reporting data just from a single reference (e.g. L102-105 in which authors use data from [32-35] reporting only data retrieved from [32]. It would be useful for the reader, especially for this particular topic to have data clearly explained. The use of Tables is encouraged of all analytes considered.

Reply: Thank you for bringing this to our attention, all errors have been corrected accordingly. Three new tables have been added to the publication.

  1. In L105-106 authors refer to a study but they do not provide reference.

Reply: Thank you very much for drawing our attention to this matter. The reference has been added accordingly.

  1. L113 authors report the median of lymphocyte count. It should be important to add the IQR for all data, where provided.

Reply: Thank you for your comment and suggestion. According to the suggestion of the Reviewer, we now include IQRs for parameters (Table 1-3).

  1. Please check the percentage reported in L 126.

Reply: The percentage reported in L126 has been corrected, thank you.

  1. I suggest to use use uniform nomeclature for the virus (Sars-CoV-2).

Reply: The nomenclature has been corrected in accordance with your suggestion, thank you.

  1. Even though most of the data are available from few days. I encourage the use of peer-reviewed papers instead of manuscript uploaded on preprint server.

Reply: In the current version of the manuscript, we have expanded the literature with new items, and some preprints have already been published in peer-reviewed journals. At the moment, preprints are a small part of the references. Please accept the list of publications in the current version.

  1. On L173, authors should refer to a ROC curve for the reported AUC.

Reply: The AUC has been corrected, thank you.

  1. Please check if all the acronyms are defined the first time they appear in the manuscript. Moreover, i would use the extended names when reported as titles of sub-headings.

Reply: The acronyms are defined when first encountered, thank you.

  1. One of the weakness of the manuscript is that data are reported carefully, but a reader would like to have a critical and in depth discussion of the data reported in literature.

Reply: Most of our article has been supplemented with new research. Thanks to this, we have also expanded the conclusions regarding individual hematological changes in COVID-19. We have also added further directions that critically refer to the research cited at work.

  1. Please check the reference used in L181-183. It seems not properly referenced. If yes,
    I suggest authors to go through the references in order to reduce inconsistencies.

Reply: Thank you for this comment. The references have been corrected.

  1. No data about cytokines is reported. I suggest to add a part related to at least IL-6.

Reply: The subsection on cytokine storm in COVID-19 has been added.

  1. Conclusions are not properly organized and is not of worth to place a Table and a figure here. Table 1 can have interest if properly placed and expanded.

Reply: Thank you for your suggestions. The conclusion has been corrected accordingly.

  1. Table caption and Conclusions are mixed together (L216-227).

Reply: It has been corrected according to your comment.

Again, thank you for your comments and suggestion.

Reviewer 2 Report

The manuscript is overall well-written. The only minor comment from me is that Figure 2 is oversimplified and does not have much a value. Please add more detail to it. For example, add the reported number of increased WBC (e.g. >10 x 10e9/L) and 2) and the reported number of decreased lymphocytes (e.g. <1 x 10e9/L) next to the cell populations.  

The conclusion is too brief and it does not carry any value. If it is there please add more detail and probably future perspectives to this paragraph.

Author Response

Bydgoszcz, June, 12 2020

Dr. Artur SĹ‚omka

Department of Pathophysiology

Ludwik Rydygier Collegium Medicum in Bydgoszcz

Nicolaus Copernicus University in Toruń

Prof. Dr. Lawrence S. Young

Editor-in-Chief

Pathogens

Dear Prof. Young,

Thank you very much for your correspondence dated on 2020-05-23 regarding our manuscript authored by Artur SĹ‚omka et al., entitled “Coronavirus disease 2019 (COVID-19): a short review on hematological  manifestations” (pathogens-818772), submitted for publication in Pathogens. We are grateful that you consider our manuscript as likely suitable for publication. We have carefully rewritten the manuscript taking advantage of the Reviewer comments.

Enclosed please find our detailed responses to specific remarks of the Referees.

Yours sincerely,

Dr. Artur SĹ‚omka

REPLY TO REVIEWER #2 COMMENTS

We would like to thank you for your insightful and valuable comments which have greatly helped us to improve the quality of the manuscript. All the changes are highlighted in yellow in the revised text.

Below, we enclose a point-by-point response to your comments.

  1. The manuscript is overall well-written. The only minor comment from me is that Figure 2 is oversimplified and does not have much a value. Please add more detail to it. For example, add the reported number of increased WBC (e.g. >10 x 10e9/L) and 2) and the reported number of decreased lymphocytes (e.g. <1 x 10e9/L) next to the cell populations.

The conclusion is too brief and it does not carry any value. If it is there please add more detail and probably future perspectives to this paragraph.

Reply: Thank you for your comments. The paper has been rewritten to describe further directions that critically refer to the research cited at work. Because accurate data on the number of cells and other parameters are presented in Tables 1-3, we would be grateful if you accept Figure 1 in the current version.

Again, thank you for your comments and suggestion.

Round 2

Reviewer 1 Report

for clarity in table I, test type should be provided.